# Association of a *CHEK2* somatic variant with tumor microenvironment calprotectin expression predicts platinum resistance in a small cohort of ovarian carcinoma

Izabela Ferreira Gontijo de Amorim[1,2°], Carolina Pereira de Souza Melo[1°],
Ramon de Alencar Pereira[1°], Sidnéa Macioci Cunha[3,4], Thalía Rodrigues de Souza Zózimo[1],
Fábio Ribeiro Queiroz[1], Iago de Oliveira Peixoto[1,5], Luciana Maria Silva Lopes[5],
Laurence Rodrigues do Amaral[6], Matheus de Souza Gomes[6], Juliana Almeida Oliveira[2,7],
Eduardo Batista Cândido[2,4], Paulo Guilherme de Oliveira Salles[1,3], Letícia da Conceição Braga[1]*

1 Laboratório de Pesquisa Translacional em Oncologia, Núcleo de Ensino, Pesquisa e Inovação, Instituto Mário Penna, Belo Horizonte, Minas Gerais, Brazil, 2 Curso de Medicina, Faculdade de Minas-FAMINAS, Belo Horizonte, Minas Gerais, Brazil, 3 Hospital Luxemburgo, Instituto Mário Penna, Belo Horizonte, Minas Gerais, Brazil, 4 Programa de Pós-graduação em Saúde da Mulher, Departamento de Ginecologia e Obstetrícia, Universidade Federal de Minas Gerais, Belo Horizonte, Minas Gerais, Brazil, 5 Programa de Pós-graduação em Biotecnologia, Fundação Ezequiel Dias-FUNED, Belo Horizonte, Minas Gerais, Brazil, 6 Laboratório de Bioinformática e Análises Moleculares, Universidade Federal de Uberlândia, Campus Patos de Minas, Uberlândia, Minas Gerais, Brazil, 7 Programa de Pós-Graduação em Ciências Aplicadas à Cirurgia e à Oftalmologia, Faculdade de Medicina, Universidade Federal de Minas Gerais, Belo Horizonte, Minas Gerais, Brazil

☯ These authors contributed equally to this work.
* leticia.braga@mariopenna.org.br

## Abstract

High-grade serous ovarian cancer (HGSOC) low overall survival rate is often attributed to platinum resistance. Recent studies suggest that the tumor associated-microenvironment (TME) is a determining factor in malignant tumor progression and DNA damage plays a crucial role in this process. Here, we sought to identify platinum resistance biomarkers associating the TME immune profile and the mutational landscape of the homologous repair pathway genes with the HGSOC patients prognosis and response to chemotherapy. Using a decision tree classifier approach, we found that platinum resistant (PR) patients were characterized by the presence of a novel deep intronic variant, the *CHEK2* c.319+3943A>T, and higher L1 expression (p = 0.016), (100% accuracy). Chek2 protein is an important component of DNA repair and L1, also known as calprotectin, is one component of the neutrophil extracellular traps (NETs) during inflammation, previously suggested as a key contributor to the metastatic process in HGSOC. Also, PD-L2 levels were significantly higher in PR patients positive for this *CHEK2* variant (p = 0.048), underscoring the need to explore its potential therapeutic role for this cancer. Our results suggest an interplay between TME and DNA repair variants that results in a multifactorial nature of HGSOC resistance to platinum chemotherapy.

**Data availability statement:** All relevant data are within the manuscript and its supporting information files.

**Funding:** This study was funded by the Ministry of Health (PRONON – Grants numbers: NUP:25000.079266/2015-09 and 25000.020618/2019-35; www.gov.br/saude/pt-br/acesso-a-informacao/acoes-e-programas/pronon-e-pronas-pcd) and Fundação de Amparo a Pesquisa no Estado de Minas Gerais - FAPEMIG (APQ-03223-17; www.fapemig.br/pt/). The funders had no role in study design, data collection and analysis, decision to publish, or preparation of the manuscript.

**Competing interests:** The authors have declared that no competing interests exist.

## Introduction

Ovarian cancer (OC) affects approximately 230,000 women worldwide per year. The five-year overall survival rate (46-45%), is associated with a high recurrence rate, leading to approximately 150,000 deaths annually. Although OC is the seventh most common cancer among women, it ranks fifth in terms of mortality [1] being the deadliest gynecologic cancer [2–4]. It is a heterogeneous disease with different classifications related to morphological and molecular characteristics, encompassing a wide range of biological features. Among these subtypes, epithelial ovarian cancers (EOCs) account for 90% of all OC cases, and high-grade serous ovarian carcinoma (HGSOC) represents the most prevalent and aggressive subtype [1].

Due to the absence of specific symptoms or effective screening tests, over 75% of OC cases are diagnosed at an advanced stage, characterized by extensive metastasis [2,4–7]. In an attempt to reduce mortality rates, significant endeavors have been dedicated to improve the early OC diagnosis and to develop novel therapeutic approaches. Nonetheless, the cure rate for OC has not demonstrated substantial improvement over the course of several decades [8,9].

The challenge inherent in discovering a cure may be attributed to the unfamiliar mechanisms underlying ovarian cancer [10]. Recent studies have shown that the parenchymal tumor cells and their non-parenchymal components change with the course of the disease [11,12], suggesting that the tumor associated-microenvironment (TME) might be a determinant factor that drives malignant tumors progression. Furthermore, TME suppress immune response against the tumor and counterbalance anti-tumor immunity [8]. DNA damage also plays a crucial role in this process, either by triggering inflammatory pathways or by generating neoantigens which, when presented by MHC molecules at the cell surface, have the potential to elicit a strong T-cell response [13]. Furthermore, genetic variants in DNA repair pathways, such as the homologous recombination repair (HR), are intrinsically connected with tumor immune profile.

HR pathway is capable of repairing DNA double-strand breaks (DSBs), being essential for genomic stability maintenance, and can arise from various sources, including endogenous cellular processes and exogenous genotoxic stress [14]. HR ensures accurate repair of DSBs through an undamaged homologous DNA sequence template to guide the repair process. This pathway is especially vital during the S and G2 phases of the cell cycle, when a sister chromatid is readily available as a template for repair [15]. Considering that platinum agents, such as cisplatin and carboplatin, exert their anti-cancer effects by forming DNA adducts that result in DNA crosslinking and subsequent DNA damage, HR pathway plays a significant role in the response to platinum-based chemotherapy (Pt-C), which is the first line therapy in OC treatment. While in normal cells, the HR pathway efficiently repairs these DNA lesions ensuring cell survival, in the context of OC, HR pathway proficiency can lead to platinum resistance [16]. Thus, besides being platinum-sensitive, HR-deficient tumors are associated to higher loads of neoantigens, resulting in increased PD-1/PD-L1 expression in immune cells surrounding TME. Previous studies have shown that PD-L1 and PD-L2 are both predictive biomarkers for the response to anti-PD-1 therapy and are also prognostic factors for several cancer types [17,18]. For this direct association with immune cells activation, both, genomic instability and critical components of the TME, such as tumor-infiltrating lymphocytes (TIL), tumor-associated macrophages (TAM) and tumor-associated tissue eosinophilias, may serve as biomarkers of tumor evolution and patient's response to treatments using immunotherapy [12,19–21].

In this sense, the current study sought to identify the association between HGSOC microenvironment and HR pathway genes variants with Pt-C response and patient prognosis. Our results may give further insights about the relationship between genomically unstable cancers and the tumor immune clearance escape. Furthermore, our results may contribute to

the identification of possible biomarkers for immunotherapy eligibility and to develop more effective HGSOC treatments.

## Materials and methods

### Patients and tissue samples

This retrospective cohort study was carried out at Mário Penna Institute - Luxemburgo Hospital - with approval from the institutional ethics committee (CAAE: 365587201.1.1001.5149) and informed written consent from all participants. For this, the medical records of 52 patients diagnosed with ovarian cancer between 2015 and 2019 were initially reviewed. After applying the inclusion and exclusion criteria, 24 patients with high-grade serous carcinoma were selected (S1 Fig). The inclusion criteria for the study were: patients diagnosed with histologically confirmed HGSOC [Federation International of Gynecology and Obstetrics (FIGO) stage I–IV] and who underwent primary debulking surgery (PDS) followed by Pt-C as first-line chemotherapy or neoadjuvant chemotherapy, when optimal PDS was not feasible, followed by interval debulking surgery. The patients were segregated considering the platinum-free interval (PFI), defined as the time between the last dose of first-line chemotherapy to the date of first recurrence. Patients were classified as either platinum-sensitive (PS) if they presented symptom relief, normalization of CA-125 levels, and absence of suspicious malignant tumor images six months after the end of Pt-C in two consecutive quarters; or platinum-resistant (PR), when disease progression occurred in an interval between two and six months or in cases where this interval was lower than two months or even no response was observed during the Pt-C (called platinum-refractory). Finally, twenty-four epithelial ovarian cancer samples from HGSOC patients were obtained, prior to any intervention or treatment, and verified by the same pathologist.

### Immunohistochemistry (IHC) assay

Ovarian biopsies were fixed in 10% buffered formalin, processed into paraffin blocks, and stored at room temperature. The IHC protocol employed an indirect HRP polymer method [22]. Each slide received diluted primary antibody (S1 Table) during primary antibody incubation. The signal amplification process involved the HiDef Detection HRP Polymer System kit in a two-stage process with sequential incubation steps, first with the HiDef Detection™ Amplifier (Mouse and Rabbit), followed by the HiDef Detection™ HRP Polymer Detector (Cell Marque Cat#951D-10, Rocklin, CA). Negative controls without primary antibody were prepared for each sample.

**Interpretation of CD4+, CD8+, CD68+, L1/MAC387+, PD-L1 and PD-L2 expression.** IHC sections stained for CD8+, CD4+, CD68+ e L1/MAC 387+ were viewed with a 40× objective with trinocular microscope Nikon Eclipse (Nikon®) and 20 random fields were scanned with a Nikon micro-Camera DS-Ri2 (Tokyo, Japan). For image analysis the freeware ImageJ v1.33 as well as the counter plugin, both downloaded from the NIH website (http://rsb.info.nih.gov/ij), were used. Immunostained cells for CD8+, CD4+, CD68+ and L1/MAC387+ located in the intra-tumoral region (lymphocytes and macrophage in tumor nests having cell-to-cell contact with no intervening stroma and directly interacting with carcinoma cells) and stromal-tumor region (lymphocyte and macrophage dispersed located in the stroma between the carcinoma cells, and do not directly contacting carcinoma cells) were manually counted in all 20 fields obtained from each patient, through the ImageJ program. Quantification of positive cells was performed manually by observers blinded to all clinical information. Areas containing fibrosis, adipose tissue or necrosis were avoided.

For both PD-L1 and PD-L2 IHC, tumor tissue sections were independently examined by two researchers, and the intensity of the staining was evaluated following the supplier's

recommendation: 0, negative; 1, weakly positive; 2, moderately positive and 3, strongly positive [23,24].

### Next Generation Sequencing (NGS)

Three 10 µM thick FFPE tissue sections with at least 60% tumor cellularity were subjected to deparaffinization and DNA extraction using the EZ2 AllPrep DNA/RNA FFPE kit (QIAGEN, Antwerp, Belgium), following manufacturer's specifications. DNA concentration was assessed with the Qubit™ 3 Fluorometer using the Qubit™ 1X dsDNA High Sensitivity assay kit (Thermo Fisher Scientific, Waltham, MA, USA). DNA integrity was determined through Q-PCR using the QIAseq QuantiMIZE assay kit.

NGS was performed on tumor samples from 17 of the 24 (70.8%) patients enrolled using the Qiaseq Pan-cancer Multimodal Panel® (QIAGEN, Antwerp, Belgium). Library preparation was performed according to the manufacturer's instructions, and the final size of the libraries was evaluated on a 4200 TapeStation System (Agilent Technologies Inc, Santa Clara, CA, USA). Final library quantification was assessed through Q-PCR using the QIAseq Library Quant System (QIAGEN, Antwerp, Belgium). Sequencing was carried out using a high-output kit in the NextSeq 550 System (Illumina, Inc., San Diego, CA, USA) in a 2 × 150 bp paired-end run.

### Variant annotation

Sequencing analysis and variant annotation were performed using the QIAseq Multimodal Analysis with TMB and MSI workflow from the Biomedical Genomics Analysis plug-in of the QIAGEN CLC Genomics Server 22 software (QIAGEN, Antwerp, Belgium). Filtering and variant calling were performed using the workflow's default parameters, outlined in Supplementary Methods S1 Box. Variants passing filters detected in 14 HR pathway genes (*ATM*, *BARD1*, *BLM*, *BRCA1*, *BRCA2*, *BRIP1*, *CHEK2*, *GEN1*, *NBN*, *PALB2*, *RAD51*, *RAD51B*, *RAD51D*, and *RAD52*) and that have been described as variant of uncertain significance (VUS), benign/likely benign or pathogenic/likely pathogenic in ClinVar, were analyzed considering patients group (PR or PS) and immune profile.

### Statistical analysis

Continuous variables were expressed as means and standard deviations, and means were compared using the t-test. Categorical variables were presented as absolute numbers and percentages, and groups were compared with Fisher's exact test. Survival curve comparisons were performed using the Log-rank (Mantel-Cox) test. The relative risk for selected parameters was obtained through Cox-proportional Hazard (CoxPH) Regression using RStudio and GraphPad Prism [25,26]. Predictions of five-year overall survival using selected variables were performed using adjusted survival curves for the Cox Proportional Hazards (CoxPH) Model. The metrics time-dependent Brier score, time-dependent receiver operating characteristic (ROC) curve with their respective Area under the Curve (AUC) and a calibration plot were used to assess each model performance. Specific RStudio packages were used for these analyses, as described in Supplementary Methods S2 Box.

To identify among all the variants detected those able to distinguish PR from PS patients, we used a decision tree algorithm to classify the complete dataset utilizing the WEKA software (Waikato Environment for Knowledge Analysis, version 3.6.11, University of Waikato New Zeland). Subsequently, we conducted Leave-One-Out Cross-Validation (LOOCV) to estimate classification accuracy and evaluate model generalization.

All data utilized for the analyses mentioned above are available in S2 Table.

## Results

### Clinical patient characteristics

The median age at diagnosis was 58 years and almost three-quarters of the cases (70.8%) were considered PS. The clinicopathological characteristics of recruited HGSOC patients are summarized in the S3 Table.

### Relationship of CD4, CD8, CD68, L1/MAC387 with platinum response, overall survival, and primary debulking surgery

IHC analysis was performed on the tissue sections for identification of T-helper (CD4) and cytotoxic T lymphocytes (CD8), and of tumor-associated (CD68; resident macrophage) and M1-polarized macrophages (L1/MAC387) (Fig 1). The amount of infiltrating CD4+, CD8+, CD68+, and L1/MAC387 (sum of 20 fields) was quite different between HGSOC patients inside both, PR and PS, groups (S2 Fig and S4 Table). Nevertheless, intergroup comparison did not reveal any significant difference.

An unbalanced TME can induce both beneficial and adverse consequences for tumorigenesis. For this reason, we evaluated the ratios between tumor-infiltrating lymphocyte subtypes and tumor-associated macrophages, proposing the following relationships: cytotoxic and helper T lymphocytes (CD8+ T cell/ CD4+ T cell), cytotoxic T lymphocyte and M0 macrophages (CD8+ T cells/ CD68+) and M0 and M1-polarized macrophages (CD68+/ L1.MAC387+). No differences were observed between the median of CD8+/CD4+, CD8+/ CD68+, and CD68+/L1.MAC387+ - cells ratios compared among groups regarding platinum sensitivity, 5-year overall survival, and primary debulking surgery (PDS) (S4 Table).

### IHC staining of PD-L1 and PD-L2 in HGSOC tissues

The PD-L1 and PD-L2 expression profiles were obtained through semi-quantitative assessment based on scores. IHC analysis showed that the positive expressions of PD-L1 and PD-L2

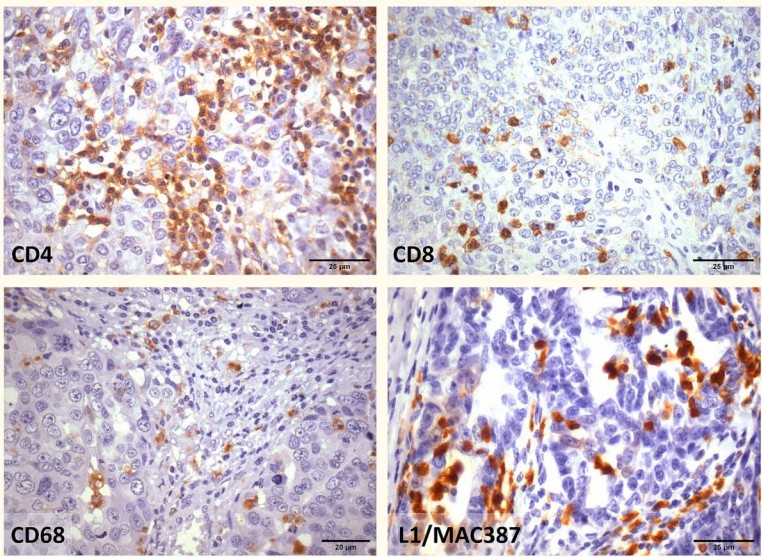

**Fig 1. Photographs showing representative immunohistochemical staining patterns of CD4, CD8, CD68, and L1/MAC387 in HGSOC cells.** Original magnification: High magnification, × 400. Scale bar: 25 μm, magnification, × 400.

were mainly detected in the cytoplasm of HGSOC cells (Fig 2a). A differentiated profile was observed between PD-L1 and PD-L2 in PS and PR groups. While the PD-L1 expression was more homogeneous in the PS and PR groups, PD-L2 expression appeared to be stronger in the PR group (86% of patients exhibited moderate to intense expression) (Fig 2b). A similar profile was observed for the association of death and debulking with PD-L1 and PD-L2 expression.

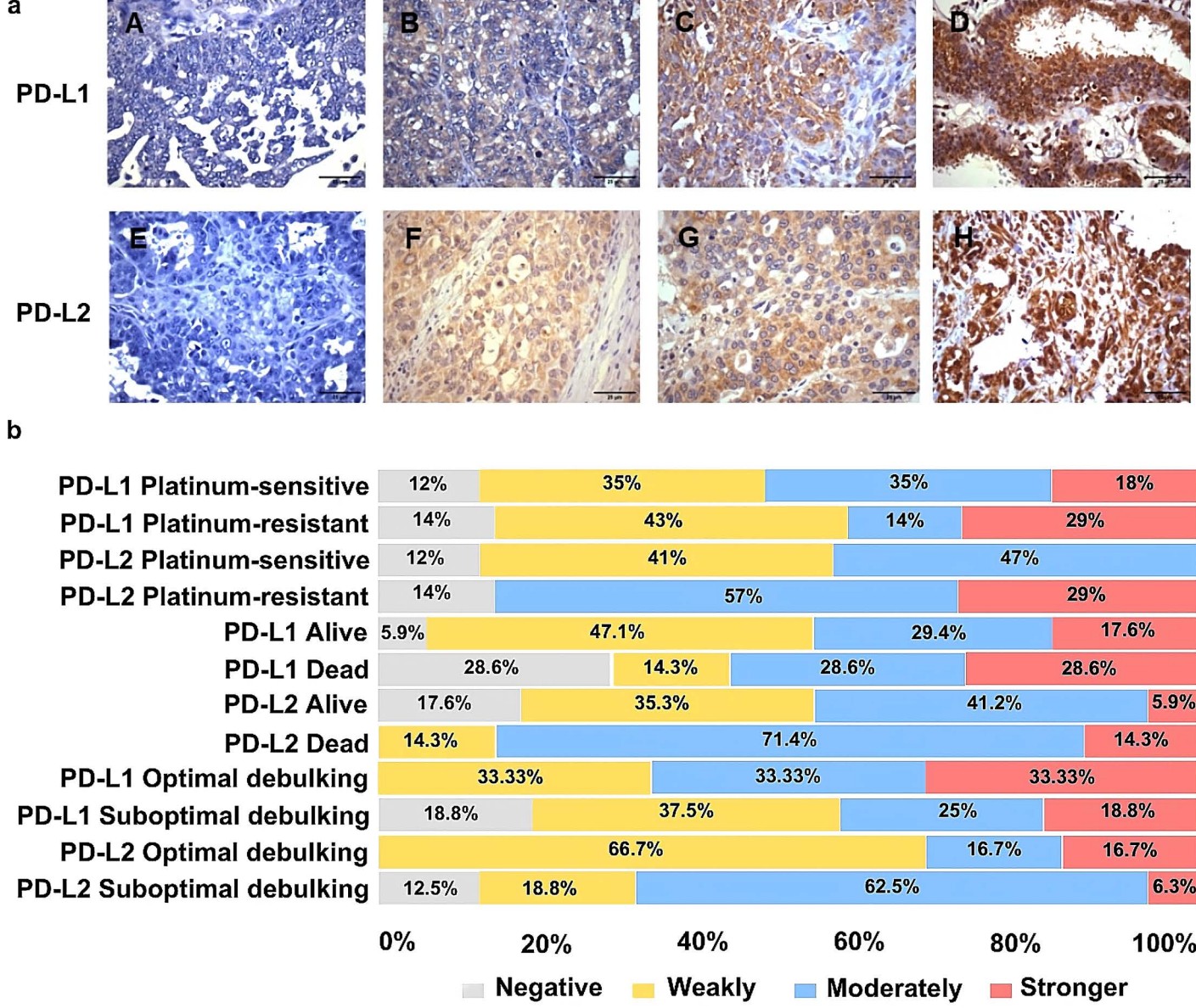

**Fig 2. Representative immunohistochemical staining patterns of PD-L1 and PD-L2 in HGSOC cells and distribution regarding patient's platinum resistance, overall survival, and primary debulking surgery *status*.** In Fig 2a, the expression of PD-L1 and PD-L2 was categorized as: (A) 0 - negative expression; (B) 1+ weakly positive expression; (C) 2+ moderately positive expression; and (D) 3+ stronger positive expression. Original magnification: A-H, x 400; Scale bar: 25µm, magnification. In Fig 2b, the expression profile of PD-L1 and PD-L2 is presented regarding the patient's platinum resistance, overall survival and primary debulking surgery status.

## HR pathway genes variant analysis

Of the 24 patients who underwent this study, 17 FFPE HGSOC samples passed on sequencing quality filters and were included in the variant analysis. A total of 128 variants were annotated in the 14 HR pathway genes analyzed. Three genes had only intronic variants: *CHEK2*, *RAD51* and *RAD52* (S3 Fig). According to ClinVar database, 52.3% (67/128) of the variants were classified as benign/likely benign, 4.7% (6/128) as oncogenic/likely oncogenic and 7.8% (10/128) as VUS or with conflicting interpretations of pathogenicity (S5 Table). Forty-five (35.1%) variants were not previously described (new), consisting of twenty-three (51.1%) intronic, nine (20%) indel, eight (17,77%) missense, three (6,66%) synonymous, and two (4.44%) nonsense. More than 50% of the detected variants were concentrated in only four genes, *RAD51B*, *BRCA2*, *BRCA1* and *RAD51*. Although all the pathogenic variants detected were identified in PS patients, no association was found between their presence and the immune markers expression (CD8, CD4, CD68 and L1) measured by the IHC assay (S4 Fig). No significant difference was observed in variant frequency distribution per gene between PR and PS groups (Fig 3).

## A deep intronic *CHEK2* variant taken together with L1 expression highlighted to classify response to platinum-based treatment

To identify between all the variants annotated and the immunological markers measured through IHC staining those with stronger association with patients response to platinum a decision tree analysis was performed. As a result, two variables were selected as relevant for patients' classification according to Pt-C resistance (Fig 4). A deep-intronic variant in *CHEK2* (CHEK2-DIV), the c.319+3943 A > T, was present in all the five PR patients and absent in 66,7% (8/12) of PS patients, significantly associating with platinum resistance (p = 0.029). Moreover, PR patients harboring CHEK2-DIV expressed significantly higher levels of L1 (p = 0.016). The cutoff point for L1 level was 84 positive cells/field. Using these two attributes the model correctly classified 100% (17/17) of the patients using the full dataset, reaching 82,4% (14/17) of accuracy in the LOOCV.

Comparing the expression of the other immunological markers between PR and PS patients positive for the CHEK2-DIV, we observed that, besides L1, PD-L2 levels were significantly higher in the PR group (p = 0.048) (S5 Fig). CD4, CD8, CD68 and PD-L1 did not differ significantly between both groups.

## Survival analysis

As expected, PtC-Resistance significantly impacts patient overall survival (Fig 5a) with PR patients having six times more risk of death than PS patients (HR = 7.7, 95% CI 2.1 - 28, p = 0.002) (Fig 5b). The same trend was observed for higher PD-L2 expression (HR = 2.1, 95% CI 0.55 – 8.3, p = 0.27) (Fig 5c) and CHEK2-DIV presence (HR = 3.9, 95% CI 0.78 - 19, p = 0.098) (Fig 5d), however the difference was not statistically significant. Accordingly, Brier Score analyses throughout the patients' follow-up time confirm the higher performance of the PtC-Resistance model, followed by the CHEK2-DIV model (S6 Table and S6a Fig). The PD-L2 model presented a modest performance, close to reference. The calibration plot confirmed these results (S6b Fig). Model discriminatory power was assessed through time-dependent ROC analyses (S6 Table and S6c Fig). CHEK2-DIV model presented relatively high discriminatory power between PR and PS, mainly in the first three years after patient diagnosis. PD-L2 model significantly lost power after 2.5 years of follow-up (S7 Fig). No impact on patient overall survival was observed for the other immunological markers, including L1 (Fig 5d).

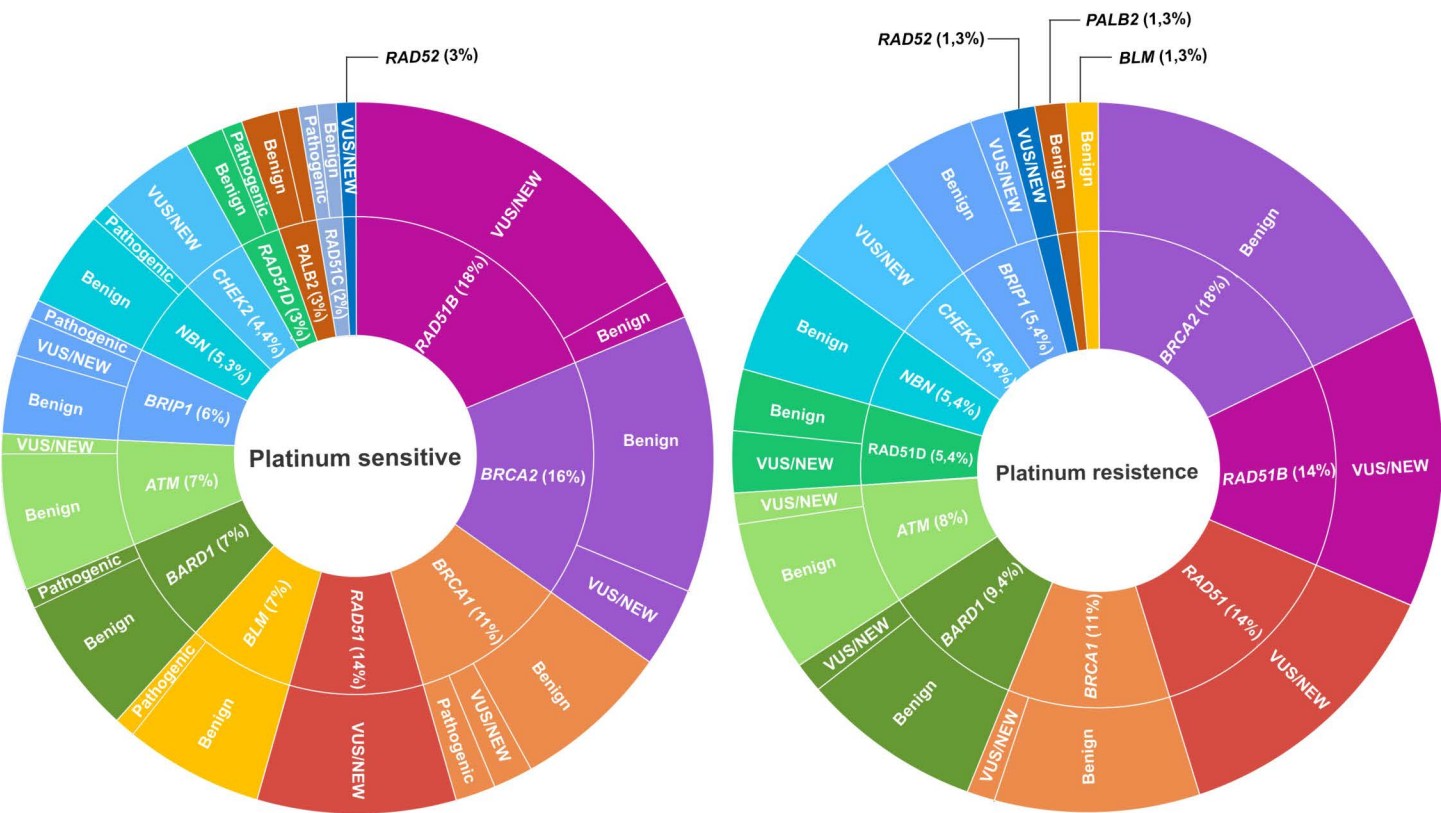

**Fig 3. Distribution of the somatic variants in HR-Related genes considering Platinum response.** In this chart, we compare the distribution of somatic variants found in genes involved in HR DNA Repair between PS (left) and PR (right) groups. Each gene is represented by a unique color. Variants are divided by genes in the inner circle and subdivided by ClinVar classification in the outer.

## Discussion

This retrospective cohort study aimed to assess the tumor microenvironment immune profile and the HR pathway mutational landscape of HGSOC patients, associated with the prognosis and chemotherapy response. Several studies emphasize the importance of TILs [27,28] and of tumor-associated monocyte-derived macrophages (TAMs) as a prognostic factor in malignancies, such as ovarian cancer [29]. At the same time, somatic missense, indels, and deep intronic variants in genes such as *BRCA1*, *BRCA2*, and others involved in HR repair have been associated with cancer and treatment resistance [30,31]. Through decision tree classifier, we selected the best attributes, among all the IHC markers measured and variants noted, for the patient's classification regarding Pt-C response. We identified that the patient's status for a novel deep intronic variant, the *CHEK2* c.319+ 3943A > T, combined with L1 expression in TME allowed the correct classification of 100% of the patients in the PR or PS groups. PR patients were characterized by CHEK2-DIV presence and higher L1 expression.

Chek2 protein plays a crucial role in the DNA repair process, participating in the regulation of various substrates, including Brca1/2, p53, and protein controlling the cell cycle dynamics [32]. Chek2 abnormal expression is one of the mechanisms driving the development of acquired resistance to DNA damage in a substantial number of clinically relevant tumors [33]. In patients harboring the CHEK2-DIV, the average variant allele frequency (VAF) was low (0.03), suggesting the existence of resistant clones at diagnosis. These clones may potentially be accountable for disease recurrence post-chemotherapy cessation. Deep

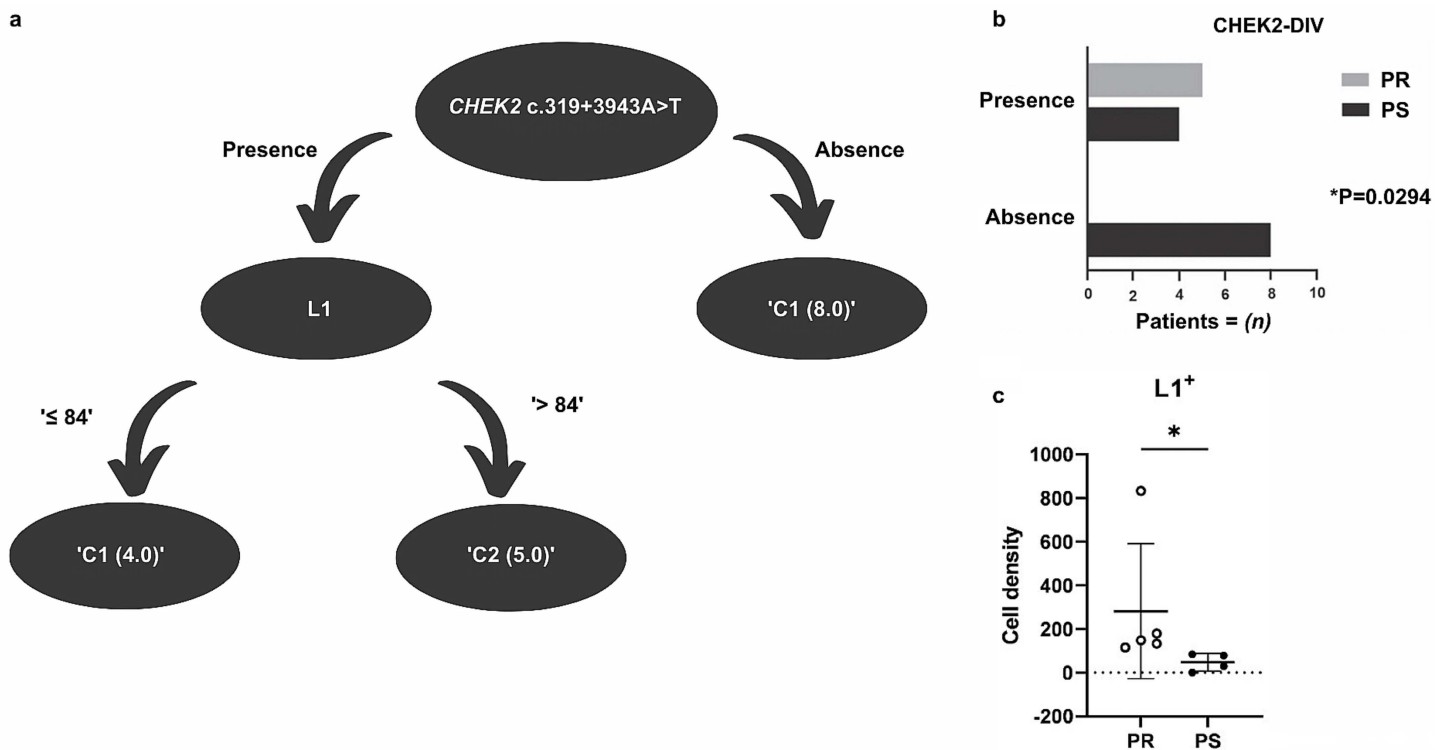

**Fig 4. Platinum-response classifier built with annotated variants and immunological markers.** (a) Supervised decision tree classifier analysis selected two variables as important for patients classification regarding PtC-response: Most PS patients were characterized by absence of the CHEK2-DIV ($n = 8$), while PR patients were characterized by CHEK2-DIV presence and higher levels of L1 ($n = 5$). (b) Coherently, CHEK2-DIV absence was significantly associated with Pt-C sensitivity ($p < 0.05$) and (c) L1 expression was significantly lower in PS patients with CHEK2-DIV.

intronic variants possess the capability to modify the consensus splicing sequence, resulting in the partial retention of introns or the activation of pseudoexons [34]. The CHEK2-DIV may encode a truncated protein due to the premature creation of a stop codon following the inclusion of pseudoexons. Subsequently, the mRNA would be degraded [35], reducing Chek2 copies within the nucleus, compounding the previously reported degradation of Chek2 protein induced by cisplatin treatment. Such reduction would hinder cell cycle control and prevent cell apoptosis, contributing to tumor chemoresistance, ultimately influencing the patient's prognosis [33,36,37]. An alternative hypothesis involved a potential gain of function associated with CHEK2-DIV. The mutated mRNA might evade degradation mechanisms and generate a protein with "non-canonical" activities, this altered protein could enhance the activation of repair proteins, such as Brca1 and Brca2, resulting in greater cellular resistance to DNA-damaging agents, such as cisplatin. Gain-of-function proteins involved in the HR pathway have been reported in association with treatment resistance, as they improve DNA repair capabilities [38].

L1, also known as calprotectin, is a heterodimer composed by two calcium-binding proteins S100A8 and S100A9, whose upregulation was reported in different cancers, increasing tumor growth, invasion and metastasis [39]. Furthermore, during inflammation the neutrophil extracellular traps (NETs) secrete calprotectin. A recent study suggested that NETs released on the omentum's surface may constitute a significant contribution to the metastatic process in HGSOC, functioning as a trap to attract tumor cells [40]. Indeed, calprotectin has been suggested as a plasma biomarker for advanced HGSOC [41]. Although not specially highlighted by the decision tree analysis, PD-L2 higher expression was also significantly

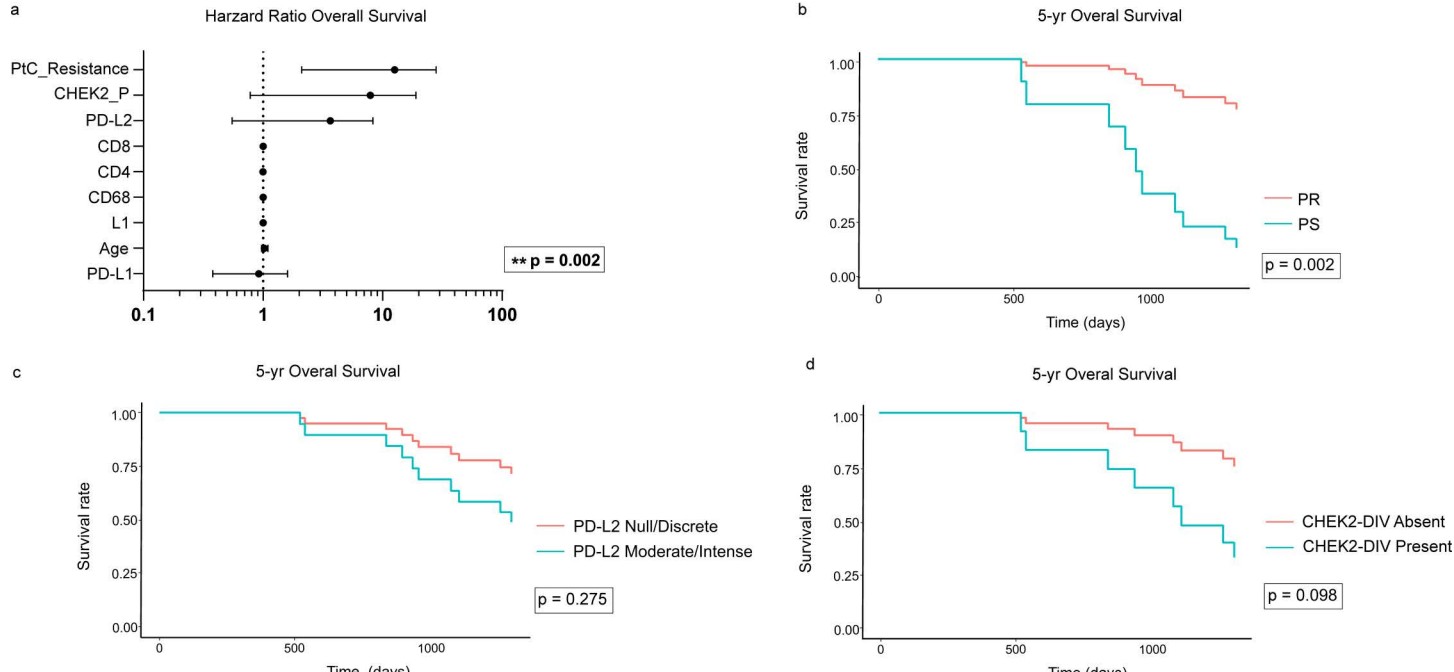

**Fig 5. Survival analysis.** (a) The CoxPH regression analysis with the markers analyzed in this study. (b-d) Adjusted survival curves for the CoxPH Model for selected variables illustrate the differences in patients' 5-year OS regarding Pt-C resistance, PD-L2 expression and CHEK2-DIV presence. Significance analyses were performed using the Wald test.

associated with the Pt-C resistance in CHEK2-DIV carriers. These two immunological markers were previously linked to a poorer prognosis in cancer patients, supporting our findings [18,41,42]. Interestingly, when considering all patients irrespective of CHEK2-DIV status, there was no significant difference in L1 (p = 0.082) or PD-L2 (p = 0.172) levels between PR and PS groups. In other words, neither are higher levels of L1 and PD-L2 alone sufficient to confer a Pt-C resistant phenotype, nor is the CHEK2-DIV presence. Rather, it seems to occur an interplay between both factors that results in the tumor response to chemotherapy.

Another noteworthy observation in our study was the PD-L1 and PD-L2 expressions in 87.5% of ovarian cancer samples analyzed. Although we found no association of PD-L1 expression with the HGSOC patients OS and PDS status, consistent with previous studies [43]. These same clinicopathological parameters showed a worsening trend in patients with moderate to intense staining for PD-L2 compared to those with null to discrete. The oversight of PD-L2 presence in tumor cells over the years has limited the availability of therapeutic options targeting this specific molecule [44]. Nevertheless, recent studies have demonstrated that a PD-L2 based immunotherapy can be a promising strategy for treating ovarian cancer [44–46].

A potential therapeutic strategy arises with the administration of pembrolizumab, an antineoplastic agent and immune checkpoint inhibitor [47] that interacts with PD1 on cell surfaces, inhibiting the PD-L1 and PD-L2 ligands. Research indicates that pembrolizumab exhibits a positive response in tumors expressing PD-L2+/PD-L1, suggesting that the PD1-PD-L2 complex may play a role independent of PD-L1 [46,48]. Additionally, combining pembrolizumab with paclitaxel, as demonstrated in the study by Lam et al [49], has emerged as a promising therapeutic approach, yielding a durable clinical response. These results underscore the need to explore the potential therapeutic role of PD-L2, both as an isolated

factor and in combination, and propose new therapeutic strategies to enhance the efficacy of immunotherapy.

Finally, in IHC analysis, we did not observe any association between, neither the absolute values of infiltrating CD4+ and CD8+ T cells nor the CD8+/CD4+ ratio, with clinical outcomes, such as optimal PDS, 5-year OS and treatment response. The heterogeneity of the TME among patients and within tumors is an inherent characteristic of HGSOC [50]. The lack of association of CD8+ cell infiltration and patients outcome have been described before [8,51,52]. On the other hand, Pinto et al. [52] found that CD4+ TILs have been independently associated with improved PFS and OS in HGSOC, after multivariate analysis adjustment. These ambiguous results could be due to the counting method used to measure these marker expressions, which may vary between different laboratories. In our methodology, we consider all the twenty fields of the sample for counting, and not uniquely the hotspot regions. It is important to highlight that the best method of assessment of this lymphocytic infiltrate in ovarian cancer has not yet been established [53].

## Conclusion

The multifactorial nature of tumor resistance to chemotherapy is well established and has been a subject of several studies in recent years. In this study, we investigated the association of genetic variants in HR pathway genes with the immune profile of the tumor microenvironment and the HGSOC patients' response to platinum. The association identified among the presence of a novel deep intronic variant, *CHEK2* c.319+3943A>T, and elevated expression of immunological markers, specially L1 and PD-L2, previously linked to poorer prognosis in PR patients, offers potential insights for identifying novel prognostic biomarkers. Moreover, our results underscore the imperative to investigate innovative therapeutic strategies that explore the potential of PD-L2 as a biomarker and to evaluate its role on HGSOC resistance more comprehensively. Despite the study's limitations, such as the small sample size, its unicentric design, and convenience sampling, its primary strength lies in its potential for improved stratification of ovarian cancer patients regarding Pt-C resistance. Additional studies with a more significant number of patients could improve and validate these results.

## Supporting information

**S1 Table. Primary antibodies used for IHC.**
(PDF)

**S2 Table. Clinical data for statistical analyses.**
(XLSX)

**S3 Table. Demographic and clinical characteristics of patients with high-grade serous ovarian cancer (HGSOC).**
(PDF)

**S4 Table. Comparison of T cell and Macrophage levels or ratios.**
(PDF)

**S5 Table. Variants annotated in the HR pathway genes analyzed.**
(XLSX)

**S6 Table. Metrics for survival model performance evaluation.**
(XLSX)

**S1 Fig. Flowchart illustrating the selection process for patients with high-grade serous ovarian cancer (HGSOC).**
(TIF)

**S2 Fig. Immunological markers expression regarding Platinum sensitivity.** Comparison of immunological markers average expression between patients platinum sensitive (PS) and resistant (PR) was performed using Mann-Whitney test. ns = not significant.
(TIF)

**S3 Fig. Distribution of variants identified in HR genes considering Platinum response.** Each variant type is associated with a specific color, as illustrated. Variant types are distributed according to the genes, highlighting the predominance of intronic variants (represented in green).
(TIF)

**S4 Fig. Immunological markers expression regarding the presence of pathogenic variants.** Comparison of immunological markers average expression between patients with and without pathogenic variants in any of the HR pathway genes screened by the Pan-Cancer panel was performed using Mann-Whitney test. ns = not significant.
(TIF)

**S5 Fig. Immunological markers expression in CHEK2-DIV carriers.** Comparison of immunological markers average expression between CHEK2-DIV patients sensitive and resistant to platinum was performed using Mann-Whitney test (a) or Fisher exact test (b). Only L1 and PD-L2 were statistically significant higher in PR patients.
(TIF)

**S6 Fig. Time-dependent metrics for survival model performance evaluation.** (a) Time-dependent Brier Score. The lower the Brier Score, the higher the model prediction capacity. (b) Calibration plot. Here, we observe that PD-L2 model is not well calibrated with few points lying in a straight line. On the other hand, PtC-Resistance model is very close to the diagonal line, which represents a perfect calibration. (c) The time-dependent AUC shows that PtC-Resistance and CHEK2-DIV models keep a good discriminatory power along the follow-up time, while PD-L2 model loses performance significantly after the first 2.5 years. The time range considered for this analysis corresponds to the time needed to occur 20% to 80% of events.
(TIF)

**S7 Fig. Time-dependent ROC curves for the survival models analyzed.** From top to bottom: CHEK2-DIV model (black line), PD-L2 model (blue line), PtC-Resistance model (red line), and the three plots merged. Here we can observe how PD-L2 model loses discriminatory power with time.
(TIF)

**S1 Box. Details of the variant calling protocol.**
(PDF)

**S2 Box. Scripts for statistical analyses using RStudio.**
(PDF)

## Author contributions

**Conceptualization:** Izabela Ferreira Gontijo de Amorim, Carolina Pereira de Souza Melo, Ramon de Alencar Pereira, Sidnéa Macioci Cunha, Eduardo Batista Cândido, Letícia da Conceição Braga.

**Data curation:** Izabela Ferreira Gontijo de Amorim, Carolina Pereira de Souza Melo, Ramon de Alencar Pereira, Sidnéa Macioci Cunha, Thalía Rodrigues de Souza Zózimo, Iago de Oliveira Peixoto, Laurence Rodrigues do Amaral, Matheus de Souza Gomes.

**Formal analysis:** Izabela Ferreira Gontijo de Amorim, Carolina Pereira de Souza Melo, Ramon de Alencar Pereira, Thalía Rodrigues de Souza Zózimo.

**Funding acquisition:** Paulo Guilherme de Oliveira Salles, Letícia da Conceição Braga.

**Investigation:** Izabela Ferreira Gontijo de Amorim, Carolina Pereira de Souza Melo, Ramon de Alencar Pereira, Sidnéa Macioci Cunha, Iago de Oliveira Peixoto.

**Methodology:** Izabela Ferreira Gontijo de Amorim, Carolina Pereira de Souza Melo, Ramon de Alencar Pereira, Thalía Rodrigues de Souza Zózimo, Fábio Ribeiro Queiroz, Laurence Rodrigues do Amaral, Matheus de Souza Gomes, Juliana Almeida Oliveira.

**Project administration:** Izabela Ferreira Gontijo de Amorim, Carolina Pereira de Souza Melo, Sidnéa Macioci Cunha, Letícia da Conceição Braga.

**Resources:** Letícia da Conceição Braga.

**Software:** Fábio Ribeiro Queiroz, Laurence Rodrigues do Amaral, Matheus de Souza Gomes.

**Supervision:** Luciana Maria Silva Lopes.

**Writing – original draft:** Izabela Ferreira Gontijo de Amorim, Carolina Pereira de Souza Melo, Thalía Rodrigues de Souza Zózimo, Juliana Almeida Oliveira.

**Writing – review & editing:** Luciana Maria Silva Lopes, Eduardo Batista Cândido, Paulo Guilherme de Oliveira Salles, Letícia da Conceição Braga.

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
