## [Decision Letter · Decision Letter 0]

15 Jul 2024

PONE-D-24-16271Association of a *CHECK2*  somatic variant with tumor microenvironment calprotectin expression predicts platinum resistance in a small cohort of ovarian carcinomaPLOS ONE

Dear Dr. Braga,

Thank you for submitting your manuscript to PLOS ONE. After careful consideration, we feel that it has merit but does not fully meet PLOS ONE’s publication criteria as it currently stands. Therefore, we invite you to submit a revised version of the manuscript that addresses the points raised during the review process.

We look forward to receiving your revised manuscript.

Kind regards,

Alvaro Galli

Academic Editor

PLOS ONE

Journal Requirements:

3. Please ensure that you include a title page within your main document. You should list all authors and all affiliations as per our author instructions and clearly indicate the corresponding author.

Reviewers' comments:

Reviewer's Responses to Questions

**Comments to the Author**

1. Is the manuscript technically sound, and do the data support the conclusions?

Reviewer #1: Partly

Reviewer #2: Yes

2. Has the statistical analysis been performed appropriately and rigorously? 

Reviewer #1: No

Reviewer #2: Yes

3. Have the authors made all data underlying the findings in their manuscript fully available?

Reviewer #1: No

Reviewer #2: Yes

4. Is the manuscript presented in an intelligible fashion and written in standard English?

Reviewer #1: Yes

Reviewer #2: Yes

5. Review Comments to the Author

Reviewer #1: The authors perform a search for biomarkers of Platinum Resistance among HGSOC patients.

Comments:

1) It is stated: Yes - all data are fully available without restriction.

Unfortunatelly, data were not included into the submission.

Please, provide the raw data, either as supplementary csv files

or through a repository, such as, e.g., the Mendely data.

2) The authors did LOOCV to assess performance of the selected two predictors.

However, they failed to embed into cross-validation framework the feature selection itself.

As a consequence, the findings may be misleading.

3) Please complement the already performed survival analysis by the predictive approach:

Use the time-dependent Brier score to quantify how well the model can predict survival.

Use time-dependent ROC to quantify how well can the model discriminate.

Check, whether the predictions of survivals are well-calibrated.

4) Please use the Adjusted survival plots based on the fitted CoxPH model,

rather than the Kaplan-Meier plots.

Reviewer #2: Major comments:

1. Lines 229-238

1.a. The authors used a decision tree classifier to identify variables and value cutoffs that discriminate between the two classes PR and PS. The classifier successfully discriminates between the two classes when using all the classes to fit the model. I believe this model is the one that produces the cutoff value of 84 for the L1 levels. I think this should be mentioned.

1.b. For the leave one out cross-validation (LOOCV). How did you choose which sample to leave out? Did you fit the 17 possible classification models? If so, is the 14/17 accuracy the best or the worse accuracy among all models?

1.c. I suggest that the authors use the term “decision tree classifier” instead of just “machine learning”.

2. Data and code availability

2.a. The authors performed analysis using RStudio. It would be beneficial if the corresponding scripts are included in the submission as supporting information, along with the necessary files for the code to run.

Minor comments:

1. There is a typo in the title. The gene name should read CHEK2 instead of CHECK2.

2. Line 168. duplicated word “Table”

3. Line 191. Choose one way to write PDL1 or PD-L1

6. PLOS authors have the option to publish the peer review history of their article (what does this mean? ). If published, this will include your full peer review and any attached files.

**Do you want your identity to be public for this peer review?** For information about this choice, including consent withdrawal, please see our Privacy Policy .

Reviewer #1: No

Reviewer #2: No

---

## [Author Response · Author response to Decision Letter 1]

5 Sep 2024

Response to Reviewers

Article title: Association of a CHEK2 Somatic Variant with Tumor Microenvironment Calprotectin Expression Predicts Platinum Resistance in a Small Cohort of Ovarian Carcinoma

Authors: Izabela Ferreira Gontijo de Amorim, Carolina Pereira de Souza Melo, Ramon de Alencar Pereira, Sidnéa Macioci Cunha, Thalía Rodrigues de Souza Zózimo, Fábio Ribeiro Queiroz, Iago de Oliveira Peixoto, Luciana Maria Silva Lopes, Laurence Rodrigues do Amaral, Matheus de Souza Gomes, Juliana Almeida Oliveira, Eduardo Batista Cândido, Paulo Guilherme de Oliveira Salles, Letícia da Conceição Braga

Date: September 5, 2024

Submission Number: PONE-D-24-16271

We would like to express our gratitude to the reviewers for their detailed and constructive comments on our manuscript. Your observations have significantly improved the quality of our work. We have comprehensively revised the manuscript to enhance the clarity and accuracy of our descriptions. In response to your specific comments, we have made the following revisions:

Journal Requirements

Answer 1: All file names have been revised to meet PLOS ONE’s requirements.

Answer 2: We are sorry for the inconvenience. We have now included the title page in into the beginning of the manuscript file.

3. Please ensure that you include a title page within your main document. You should list all authors and all affiliations as per our author instructions and clearly indicate the corresponding author.

Answer 3: We are sorry for the inconvenience. We have now included the title page in into the beginning of the manuscript file.

Review Comments to the Author

Reviewer #1

Comment 1: It is stated: Yes - all data are fully available without restrictions. Unfortunately, the data were not included in the submission. Please provide the raw data, either as supplementary CSV files or via a repository, such as Mendeley Data.

Answer 1: We apologize for this inconvenience. We have taken your comment seriously and have thoroughly revised all tables containing clinical data and other relevant datasets. These are now being submitted along with the manuscript as S2 Table, ensuring the integrity and transparency of our research. During the revision process, we found a mistake in two event/censored patients. The correction resulted in a slight change in hazard ratio values, which have been adjusted in the manuscript. The topic is Survival Analysis of the Results session.

Comment 2: The authors performed LOOCV to assess the performance of the two selected predictors. However, they failed to incorporate the feature selection itself into the cross-validation framework. Consequently, the findings may be misleading.

Answer 2: Thank you for your insightful comment. We understand the importance of embedding feature selection within the cross-validation framework to avoid potential bias. However, we would like to clarify that the selection of the two predictors was performed independently before the application of the LOOCV, and these predictors were fixed throughout the entire validation process. This means that the feature selection was not part of the LOOCV process itself but was conducted as a preliminary step. The LOOCV was then applied to evaluate the performance of the classifiers using these pre-selected predictors. Therefore, there was no need to include the feature selection within each fold of the LOOCV, as the attributes used in the model were already determined in advance. This approach ensures that the model's performance is assessed based on the chosen predictors without any additional feature selection bias during the validation phase. We hope this clarifies our methodology and are open to further discussion if necessary.

Comment 3: Please complement the survival analysis already performed by the predictive approach: Use the time-dependent Brier score to quantify how well the model can predict survival. Use time-dependent ROC to quantify how well the model can discriminate. Check whether the survival predictions are well-calibrated.

Answer 3: Thank you for pointing out these complementary analyses. They are now presented in Supplementary Files (S6 Table, and S5 and S6 Figures). A brief citation was also included in the Statistical Analysis topic of the Material and Methods session and in the Survival Analysis topic of the Results session.

Comment 4: Please use the survival plots adjusted based on the adjusted CoxPH model, instead of the Kaplan-Meier plots.

Answer 4: As recommended, the Kaplan-Meier plots (Figure 5 b-d) were substituted by correspondent adjusted survival curves for the CoxPH Model. A correction was also made in the Statistical Analysis topic of the Material and Methods session.

Reviewer #2

Comment 1a: Lines 229-238. The authors used a decision tree classifier to identify variables and cutoffs that discriminate between the two classes PR and PS. The classifier successfully discriminates between the two classes when using all classes to fit the model. I believe this model is the one that produces the cutoff of 84 for L1 levels. I think this should be mentioned.

Answer 1a: Thank you for your suggestion. This information was inserted in the Results session, line 286.

Comment 1b: For leave one out cross validation (LOOCV). How did you choose which sample to leave out? Did you fit all 17 possible classification models? If so, is the 14/17 accuracy the best or worst accuracy among all models?

Answer 1b: We appreciate your question regarding the LOOCV process. In the LOOCV approach, each of the 17 samples was systematically left out once, and the model was trained on the remaining 16 samples. This process was repeated 17 times, leaving out a different sample each time. The accuracy of 14/17 reflects the number of correct predictions out of the 17 folds. To clarify, this accuracy is calculated by averaging the performance across all folds rather than representing the performance of a single model. Therefore, it reflects the overall generalizability of the model across the entire dataset rather than being the best or worst model accuracy. We will ensure this explanation is more clearly articulated in the revised manuscript.

Comment 1c: I suggest the authors use the term “decision tree classifier” instead of just “machine learning”.

Answer 1c: Thank you for your suggestion. The text became clearer and more focused with this change. The revisions have been made on lines 49, 285, and 315.

Comment 2a: Data and code availability. The authors performed the analysis using RStudio. It would be helpful if the corresponding scripts were included in the submission as supporting information, along with the files needed to run the code.

Answer 2a: All scripts used in RStudio are described in Supplementary Methods Box 2.

Comment 3: There is a typo in the title; the gene name should be CHEK2 instead of CHECK2. Line 168, duplicate word “Table” and line 191, and please choose a way to spell PDL1 or PD-L1.

Answer 3: Thank you for your careful review of our writing. All suggestions have been accepted and successfully corrected. You can track the changes in the following lines:

• a. Corrected the gene name in the title.

• b. Deleted one of the duplicate words, "Table," on line 212.

• c. We apologize for the inconsistency in referring to the biomarker. The correct form is PD-L1, which has been corrected throughout the text.

We hope we have addressed correctly all the requirements and questions.

Kind regards,

Letícia Braga

---

## [Decision Letter · Decision Letter 1]

14 Oct 2024

PONE-D-24-16271R1Association of a *CHEK2*  somatic variant with tumor microenvironment calprotectin expression predicts platinum resistance in a small cohort of ovarian carcinomaPLOS ONE

Dear Dr. Braga,

Thank you for submitting your manuscript to PLOS ONE. After careful consideration, we feel that it has merit but does not fully meet PLOS ONE’s publication criteria as it currently stands. Therefore, we invite you to submit a revised version of the manuscript that addresses the points raised during the review process. Please submit your revised manuscript by Nov 28 2024 11:59PM. If you will need more time than this to complete your revisions, please reply to this message or contact the journal office at plosone@plos.org . Please include the following items when submitting your revised manuscript:

We look forward to receiving your revised manuscript.

Kind regards,

Alvaro Galli

Academic Editor

PLOS ONE

**Journal Requirements:**

Reviewers' comments:

Reviewer's Responses to Questions

**Comments to the Author**

1. If the authors have adequately addressed your comments raised in a previous round of review and you feel that this manuscript is now acceptable for publication, you may indicate that here to bypass the “Comments to the Author” section, enter your conflict of interest statement in the “Confidential to Editor” section, and submit your "Accept" recommendation.

Reviewer #3: (No Response)

2. Is the manuscript technically sound, and do the data support the conclusions?

Reviewer #3: Partly

3. Has the statistical analysis been performed appropriately and rigorously? 

Reviewer #3: Yes

4. Have the authors made all data underlying the findings in their manuscript fully available?

Reviewer #3: Yes

5. Is the manuscript presented in an intelligible fashion and written in standard English?

Reviewer #3: Yes

6. Review Comments to the Author

**Reviewer #3:**  The manuscript is interesting and covers the contemporary problem of predicting chemo-resistance in HGSOC. The main weakness of the study is its low patients number. My main point of interest is how did Authors choose the tumors for the study. I realize that all of them were HGSOC, but there are several subtypes according to the immunoreactivity of the host against tumor. The immunological composition of all the tumors studied seems to be quite uniform in its TIL's pattern. Could Authors explain this?

7. PLOS authors have the option to publish the peer review history of their article (what does this mean? ). If published, this will include your full peer review and any attached files.

**Do you want your identity to be public for this peer review?** For information about this choice, including consent withdrawal, please see our Privacy Policy .

Reviewer #3: No

---

## [Author Response · Author response to Decision Letter 2]

2 Nov 2024

Review Comments to the Author

Reviewer #3: The manuscript is interesting and covers the contemporary problem of predicting chemo-resistance in HGSOC. The main weakness of the study is its low patients number. My main point of interest is how did Authors choose the tumors for the study. I realize that all of them were HGSOC, but there are several subtypes according to the immunoreactivity of the host against tumor. The immunological composition of all the tumors studied seems to be quite uniform in its TIL's pattern. Could Authors explain this?

Thank you for your suggestion. This is a retrospective study that initially reviewed the medical records of 52 patients diagnosed with ovarian cancer treated at the Mário Penna Institute between 2015 and 2019. Of these, 24 patients were included in the study for meeting the inclusion criteria: confirmed diagnosis of high-grade serous carcinoma (HGSOC) with a pathological report, as well as complete clinical and follow-up data. The remaining 28 patients were excluded for the following reasons: 16 had histology different from HGSOC; 8 underwent surgery at another institution, resulting in loss of follow-up; and 4 had inadequate biological material (slides and paraffin blocks) for the necessary processing. To provide a clear understanding of the criteria applied, we included a flowchart in the Supplementary Files (S1_Fig), detailing the selection process for the patients with HGSOC included in this study. To clarify this modification, we added the following text in the “Patients and tissue samples” section – lines “117-120” of the Materials and Methods: “For this, the medical records of 52 patients diagnosed with ovarian cancer between 2015 and 2019 were initially reviewed. After applying the inclusion and exclusion criteria, 24 patients with high-grade serous carcinoma were selected”

Despite the apparent uniformity in the pattern of TILs, patients exhibited different distribution and quantification patterns. For instance, some showed a diffuse profile with a higher infiltration rate compared to others. However, due to the small sample size, it was not possible to identify or classify a significant difference between the groups or patients. This is evident in the results presented in graph (S2_Fig). We acknowledge that the sample size is relatively small, limiting the generalizability of our findings. However, with an increased sample size, the apparent uniformity of the infiltration may not be sustained. Besides that, exploratory studies are essential for generating hypotheses. Additional studies with a larger number of patients and using other molecular biology techniques may provide more robust validation of these results. It is also important to consider the significant presence of several variants, including indels, intronic, missense and nonsense. In the context of cancer genomes, they can lead to altered or truncated proteins. When these altered proteins are processed by the proteasome, and the peptide fragments are presented by HLA molecules on the surface of tumor cells, these peptides can act as neoantigens, making them potential targets for the immune response. Although the presence of neoantigens may explain the occurrence of TILs in the majority of tissue samples, it is important to note that TIL analysis alone is insufficient to demonstrate the immunoheterogeneity of HGSOC. To address these limitations more clearly and highlight the study's significant contributions, we have revised the ‘Conclusion section’. We now explicitly state, “Despite the study's limitations, such as the small sample size, its unicentric design, and convenience sampling, its primary strength lies in its potential for improved stratification of ovarian cancer patients regarding Pt-C resistance. Additional studies with a more significant number of patients could improve and validate these results.”

We hope we have addressed correctly all the requirements and questions.

Kind regards,

Letícia Braga

---

## [Editor Report · Decision Letter 2]

27 Nov 2024

Association of a *CHEK2*  somatic variant with tumor microenvironment calprotectin expression predicts platinum resistance in a small cohort of ovarian carcinoma

PONE-D-24-16271R2

Dear Dr. Braga,

We’re pleased to inform you that your manuscript has been judged scientifically suitable for publication and will be formally accepted for publication once it meets all outstanding technical requirements.

Kind regards,

Avaniyapuram Kannan Murugan, M.Phil., Ph.D.

Academic Editor

PLOS ONE
---

## [Editor Report · Acceptance letter]

PONE-D-24-16271R2

PLOS ONE

Dear Dr. Braga,

I'm pleased to inform you that your manuscript has been deemed suitable for publication in PLOS ONE. Congratulations! Your manuscript is now being handed over to our production team.

Kind regards,

on behalf of

Dr. Avaniyapuram Kannan Murugan

Academic Editor

PLOS ONE